# Presenting Information Closer to Mobile Crane Operators' Line of Sight: Designing and Evaluating Visualization Concepts Based on Transparent Displays

**Taufik Akbar Sitompul**[*]
Mälardalen University
CrossControl

**Rikard Lindell**[†]
Mälardalen University

**Markus Wallmyr**[‡]
Mälardalen University
CrossControl

**Antti Siren**[§]
Forum for Intelligent Machines

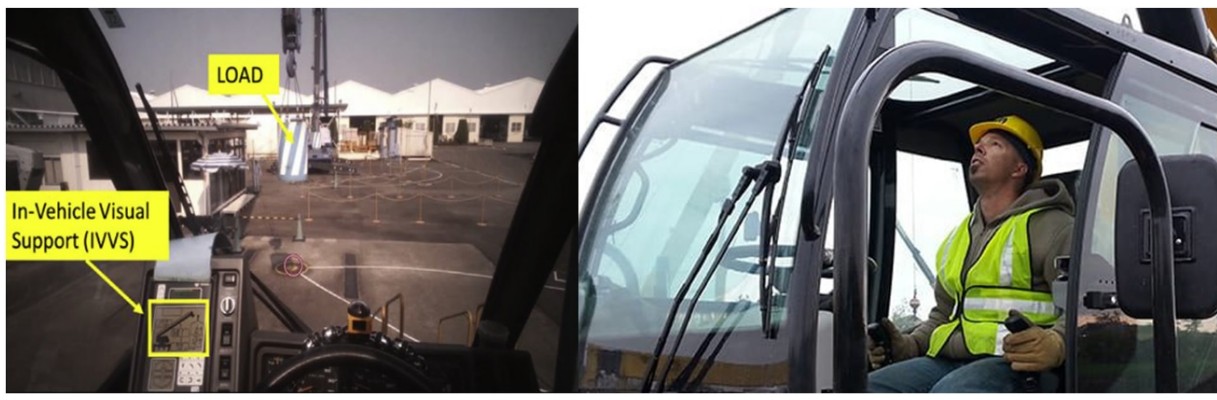

Figure 1: The left image shows an example of supportive system inside a mobile crane [6]. The right image illustrates that mobile crane operators often look at areas that are far away from the location where the supportive system is placed [2].

## ABSTRACT

We have investigated the visualization of safety information for mobile crane operations utilizing transparent displays, where the information can be presented closer to operators' line of sight with minimum obstruction on their view. The intention of the design is to help operators in acquiring supportive information provided by the machine, without requiring them to divert their attention far from operational areas. We started the design process by reviewing mobile crane safety guidelines to determine which information that operators need to know in order to perform safe operations. Using the findings from the safety guidelines review, we then conducted a design workshop to generate design ideas and visualisation concepts, as well as to delineate their appearances and behaviour based on the capability of transparent displays. We transformed the results of the workshop to a low-fidelity paper prototype, and then interviewed six mobile crane operators to obtain their feedback on the proposed concepts. The results of the study indicate that, as information will be presented closer to operators' line of sight, we need to be selective on what kind of information and how much information that should be presented to operators. However, all the operators appreciated having information presented closer to their line of sight, as an approach that has the potential to improve safety in their operations.

**Index Terms:** Human-centered computing—Visualization—Visualization application domains—Information visualization; Visualization design and evaluation methods

---

[*]e-mail: taufik.akbar.sitompul@mdh.se

[†]e-mail: rikard.lindell@mdh.se

[‡]e-mail: markus.wallmyr@crosscontrol.com

[§]e-mail: antti.siren@fima.fi

## 1 INTRODUCTION

The mobile crane is one type of heavy machinery commonly found in the construction site due to its vital role of lifting and distributing materials. Unlike tower cranes that require some preparations before they can be used, mobile cranes can be mobilized and utilized more quickly. However, mobile cranes are complex machines, as operating them requires extensive training and full concentration [10, 11]. When lifting a load, mobile cranes require wide work space in three dimensions. Operators must be cautious to prevent both the boom and the load from hitting other objects, such as structures, machines, or people. At the same time, operators must also avoid the machine from tipping over, since the machine's centre of balance is constantly changing depending on many factors, such as height and weight of the lifted load, ground's surface, and wind [19].

The complex mobile crane operation leads to operators' cognitive workload continuously high [11]. Repetitive tasks and long working hours also make operators vulnerable to fatigue and distraction, which could lower their ability to mitigate upcoming hazards. 43% of crane-related accidents between 2004 and 2010 were caused by operators [12]. In addition, mobile cranes are also considered the most dangerous machine in the construction sector, as they contributed to about 70% of all crane-related accidents [18]. Crane-related accidents can cause tremendous losses in property and life of both workers and non-workers [12, 19]. The most common crane-related accidents are electrocution due to contacts with power lines, struck by the lifted load, struck by crane parts, or a collapsing crane [18].

To assist operators, modern mobile cranes are equipped with head-down display supportive systems (see the left image in Figure 1). For example, Load Moment Indicator (LMI) systems that indicate if the maximum load capacity is approached or exceeded [19]. However, the presence of head-down displays could obstruct operators' view, and thus the information is displayed away from operators' line of sight, as shown in the right image in Figure 1. Furthermore, many

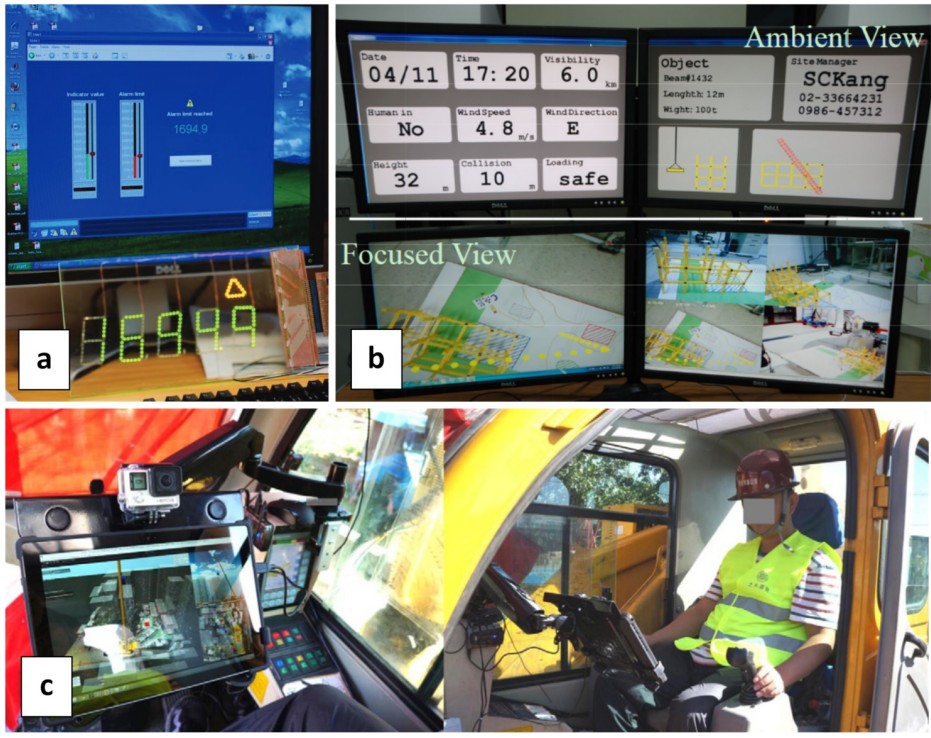

Figure 2: a. The transparent display that could be installed on the crane's windshield [14]. b. Using multiple displays for operating remote tower cranes [4]. c. Operating a mobile crane through a tablet-like display [11].

LMI systems only present numerical information that does not support operators' contextual awareness, and thus consequently requires extra cognitive effort to interpret the meaning of the information [9]. In this case, the benefit of having supportive information is nullified by both information placement and information visualization.

We hypothesize that information presented near the line of sight would benefit mobile crane operators. For example, information displayed on the windshield would allow operators to acquire the supportive information without diverting their attention from operational areas. However, this approach has its own challenges. With information presented near operators' line of sight, there is a potential risk to distract operators from their work. Therefore, the information needs to be presented cautiously, where the right information is presented at the right time, the right place, and the right intensity [8, 10]. This approach will enable operators to perform their work, while maintaining awareness of both the machine and its surroundings. For this paper, we have had the following research questions:

1. What kind of information that mobile crane operators need to know in order to perform safe operations?

2. How should the supportive information behave and look like with respect to the performed operation?

3. How do mobile crane operators perceive the proposed visualisation approach?

The rest of this paper is divided into seven sections. Section 2 reviews prior work that investigated new ways of presenting information in cranes. Section 3 explains three different activities that have been carried out to address the research questions above. Section 4 presents the visualization concepts that we have designed, while Section 5 describes the feedback that we have obtained from the operators. Section 6 describes further suggestions from the operators

and the reflection on the evaluation method used in this study. Section 6 acknowledges the limitations in this study and also outlines what could be done for future work. Section 8 finally concludes the study in this paper.

## 2 RELATED WORK

Considering the current setting where mobile crane operators receive supportive information via head-down displays, one may suggest alternative approaches using auditory or tactile modalities. However, mobile cranes are noisy and generate internal vibration due to the working engine [3] and the swinging lifted load [5]. Similarly, auditory information is already used in mobile cranes to some extent [10]. Adding more information via haptic and auditory channels could be counterproductive due to less clarity for conveying information compared to visual information [8].

Prior research indicated that visual information is still used as the primary modality for presenting supportive information in heavy machinery, including both mobile cranes and fixed-position cranes, such as tower and off-shore cranes (see Sitompul and Wallmyr [24] for the complete review). Proposed approaches for improving safety in operations using cranes also vary. For off-shore cranes, Kvalberg [14] proposed to use transparent displays that could be installed on the crane's windshield for presenting the relative load capacity (see Figure 2a), which indicates how much weight a crane can lift depending on how far and high the load will be lifted [21]. For remotely controlled tower cranes, Chi et al. [7] proposed multi-displays where each display presents certain information, such as machine status, lifting path, potential collision, and multiple views of the working environment (see Figure 2b). Fang et al. [11] proposed a tablet-like display in mobile cranes to show multiple views of the working environment, including the supportive information that indicate recommended lifting path, potential collision, and excessive load (see both images in Figure 2c).

The study of Kvalberg [14] was limited to the technical evaluation

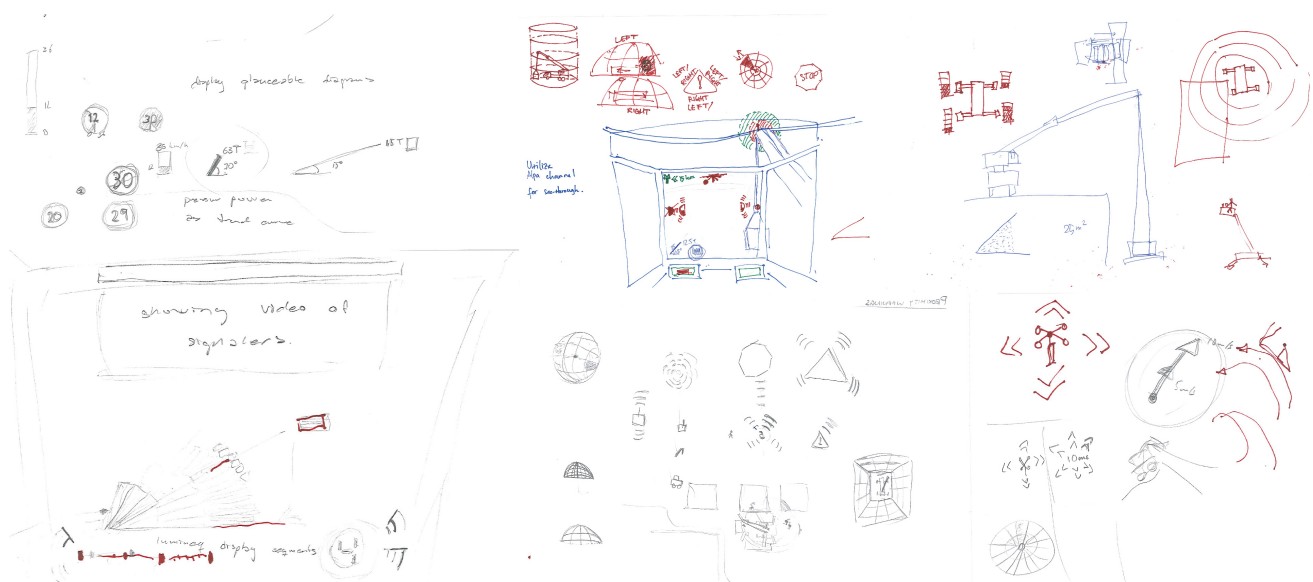

Figure 3: Some sketches that were made in our design workshop. The left sketches show how we explored the visualization of relative load capacity. The middle sketches illustrate various ways of visualizing proximity warning in different directions, distances, and height. In top right, we investigated the visualization of changes on the machine's balance. In bottom right, we sketched how the wind speed and direction should be visualized.

of the proposed transparent display (see Figure 2a). Although the experiment was carried out in a controlled environment (see Figure 2b), Chi et al. [7] involved five crane operators and 30 graduate students in their experiment. They compared the participants' performance with and without using the multi-view system. The results indicated that the participants took shorter time for completing the given task and the operator participants also perceived the multi-view system as something that could enhance safety in lifting operations. Fang et al. [11] involved five mobile crane operators in their experiment, which was also carried out in a real mobile crane. To evaluate the proposed system, they compared the performance of operators with and without the proposed system. The result showed that the operators have shorter response time and higher rate of correct responses when using the proposed system. In addition, the result from the Situation Present Assessment Method (SPAM) also indicated that the operators were able to maintain higher level of situation awareness by using the proposed system. Despite the positive result, the operators commented that the display was too small and it could also obstruct their view.

## 3  METHODS

Aligned with prior research, we hypothesize that information presented near the line of sight would benefit mobile crane operators, since they could acquire the supportive information without diverting their attention from operational areas. To address the research questions written in Section 1, three different activities were conducted, as described in the following subsections.

### 3.1  Utilizing Safety Guidelines as a Source of Information

To address the first research question and figure out which information is important for operators to perform safe operations, we reviewed four different mobile cranes operation safety guidelines [15, 17, 20, 21]. This could also have been done by asking operators or domain experts. However, this alternative may be less efficient, because operators may have different operational styles or preferences, and thus having different requirements. Furthermore, we would have missed the international aspect covered by

the guidelines from different parts of the world. Therefore, we used the safety guidelines as the starting point, as they are applicable to all operators regardless different operational styles or preferences. From the safety guidelines review, we have found that the guidelines are provided to prevent the following events:

1. Collisions that may occur between the mobile crane, its parts, or the lifted load; and nearby people, or structures at the working area. To prevent this from happening, operators should know what is around the machine and what the machine is about to do.

2. Loss of balance that could occur due to many factors, such as excessive load capacity, strong wind, or unstable ground. To avoid this event, operators should know the current state of the machine and never operate the machine beyond permitted conditions.

### 3.2  Generating Ideas Through a Design Workshop

To address the second research question, we used the findings from the safety guidelines for a design workshop to generate ideas for the appearance and behaviour of the the visualization. The workshop involved three human-computer interaction (HCI) researchers, who are also the authors of this paper. Two researchers have research expertise in human-machine interface for heavy machinery, while one researcher is a generalist.

Since the type of displays influences the form of information and how it can be presented, we firstly discussed which display is appropriate in mobile cranes. As mobile cranes have a large front windshield and operators look through it most of the time [5], the wide windshield could be used as a space for presenting the supportive information, given that the information will not obstruct operators' view. There are various commercially available displays that can be used for this purpose, such as head-mounted displays (HMDs), projection-based head-up displays (HUDs), and transparent displays. However, each of these displays has its benefit and drawback in terms of the usage in this context [24]. Using head-mounted displays, for example, Microsoft HoloLens, enables operators to see

Table 1: The profiles of mobile crane operators that we have interviewed

| No | Gender | Age | Experience | Mobile crane sizes | Knowledge about head-up displays |
|----|--------|-----|------------|--------------------|----------------------------------|
| 1 | Male | 38 years old | 12 years | 30 tons - 800 tons | Knows about it, but never tried it |
| 2 | Male | 39 years old | 20 years | 30 tons - 150 tons | Knows about it, but never tried it |
| 3 | Male | 61 years old | 38 years | 8 tons - 130 tons | Has no knowledge about it |
| 4 | Male | 53 years old | 21 years | 2.5 tons - 220 tons | Knows about it, but never tried it |
| 5 | Male | 37 years old | 7 years | 2.5 tons - 95 tons | Knows about it, but never tried it |
| 6 | Male | 45 years old | 20 years | 8 tons - 500 tons | Has tried it in a car |

the presented information exactly within their sight. Although newer HMDs come with better ergonomics, they are still not ergonomically comfortable to be used for long hours [22]. In addition, operators are already required to wear protective gears when working, and thus they may be quite reluctant to wear an additional equipment. Projection-based HUDs, like the ones available for cars, are another alternative that can be used and operators also do not need to wear another equipment. However, the quality of information presented using projection-based HUDs may be degraded in bright environments [25]. The third option is using transparent displays, like what Kvalberg [14] has used. However, this option also gives us two disadvantages [1]. Firstly, they are limited in terms of colors, since only yellow and green are currently available. Secondly, they support static visualisation only, as the display can only present information that has been specified before the display is manufactured. On the positive side, transparent displays are durable against extreme temperature, moisture, and vibration. See Figure 4 for a commercial example of transparent displays. After considering both benefit and drawback for each display, we reasoned that transparent displays are more suitable to be used in mobile cranes.

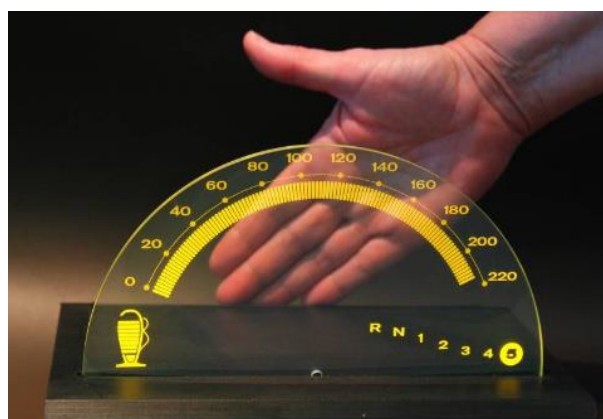

Figure 4: An example of stand-alone transparent display [1]. Each fixed element can be individually lit, whereas the widget's structure is statically rendered in the material.

We generated ideas for the visualization that could help operators in preventing hazardous situations mentioned in Subsection 3.1. We then selected some of the generated concepts based on their suitability with transparent displays (see Figure 3). Eventually, we produced eight visualisation concepts that suit the appearance, characteristics, and capability of transparent displays:

1. Two concepts for proximity warning that indicate position, distance, and height of obstacles.

2. Two concepts that indicate the balance of the machine.

3. One concept for showing wind speed and direction.

4. One concept for illustrating how much the lifted load swings.

5. One concept for presenting the relative load capacity, including the angle of the boom, the height of the hook to the ground, and the distance between the lifted load to the center of the machine.

6. A generic warning sign that tells operators to stop their current action.

The description for each visualization concept is presented in Section 4.

### 3.3 Obtaining Feedback from Mobile Crane Operators

To answer the third research question, we interviewed six mobile crane operators to validate the ease of use and possible benefits of the proposed visualizations for performing safe operations. The interviews were carried by two people, who are also the authors of this paper. After explaining both motivation and procedure of the interviews, as well as obtaining the informed consent from the operators, we collected some background information from the operators, such as age, experience as an operator, and different mobile cranes that they have used. In addition, we also asked if the operators have prior knowledge or experience on head-up displays. See Table 1 for some information about the operators that we have interviewed. Out of six operators that we interviewed, one operator has no knowledge about head-up displays. The remaining operators know about head-up displays, either through seeing commercials or driving a car that has a head-up display in it.

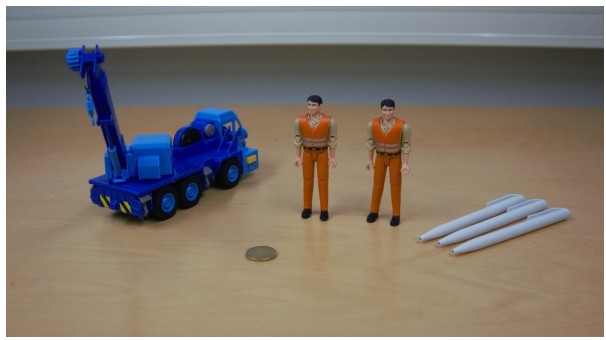

Figure 5: The tools that were used to test the operators' understanding on the proposed visualization concepts. The human toys were used to represent nearby obstacles, the coin was used to indicate where the machine's center of gravity is, and the pens were used to indicate both wind speed and direction.

After that, we presented the visualization concepts printed on papers, which illustrated how the visualizations look like in certain situations. We firstly explained what is the meaning of each component within the visualization concepts. We also used some tools (see Figure 5) to demonstrate the meaning for the visualization concepts. Once the operators confirmed that they understood the logic behind

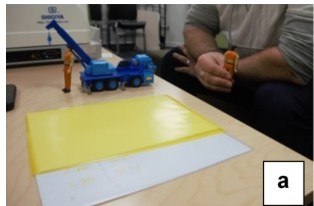 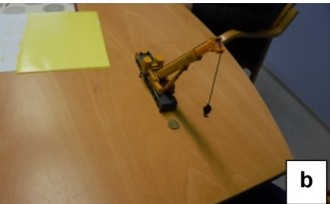 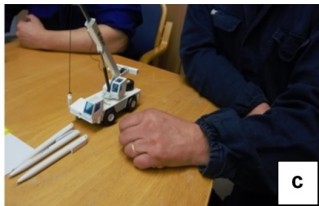 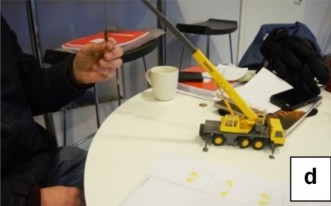

Figure 6: Some pictures that depict how we tested the operators' understanding, where the operators had to move around the provided tools according to the visualization shown on papers. a. The operators had to move the human toy(s) to the position where the obstacle is. b. The operators had to move the coin to the position where the machine's center of balance is. c. The operators had to arrange the tip of the pens to indicate the wind direction, while the number of pens represent the wind intensity; 1 pen = weak wind, 2 pens = medium wind, and 3 pens = strong wind. d. The operators were asked to move the hook that exists in the replica to show how much the lifted load is swinging.

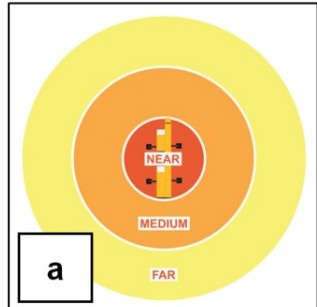 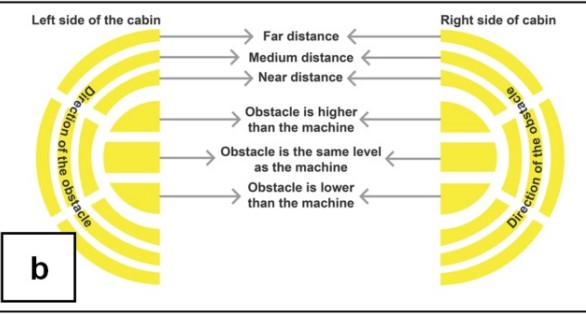 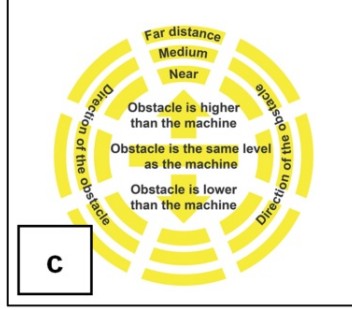

Figure 7: a. The distance between the machine and the obstacle is divided into three levels: near (1 radius), medium (2 radius), and far (3 radius). b. The meaning for each segment in the first concept of proximity warning. c. The meaning for each segment in the second concept of proximity warning.

each visualization concept, we continued with five tests which evaluated the operators' understanding on the concepts for proximity warning, balance, wind speed, swinging load, and relative load capacity. We then presented different examples of the visualization on papers to the operators. There were ten examples for each concept of proximity warning, eight examples for each concept that shows the machine's balance, eight examples for wind speed, four examples for load swinging, and eight examples for relative load capacity. Some of the examples are presented in Section 4. The test for proximity warning had increasing complexity, for example, starting from one obstacle to multiple obstacles with different heights. The operators were asked to use the provided tools, such as toys, coin, and pens (see Figure 5) and moved them according to the shown visualization (see Figure 6). This method was useful for both us and the operators, since we could understand the operators' way of thinking through their actions and the operators could show what they were thinking without having to explain everything verbally. Surprisingly, four operators had their own mobile crane replicas and we encouraged them to use their own instead. This process was repeated until all visualization concepts were described and evaluated. The generic warning sign was not evaluated, since its meaning was too obvious for the operators.

Lastly, we provided a paper that has an image of the interior view of a mobile crane's cabin. The operators were asked to place the visualization concepts, which were printed on a transparent film and then cut into pieces, on the windshield according to their preferences (see Figure 8). They were also encouraged to exclude concepts that they considered less important. Eventually, the operators were asked to describe the reasons for their decisions.

## 4 THE PROPOSED VISUALIZATION CONCEPTS

This section presents the description for each proposed visualisation concept that was generated from our design workshop.

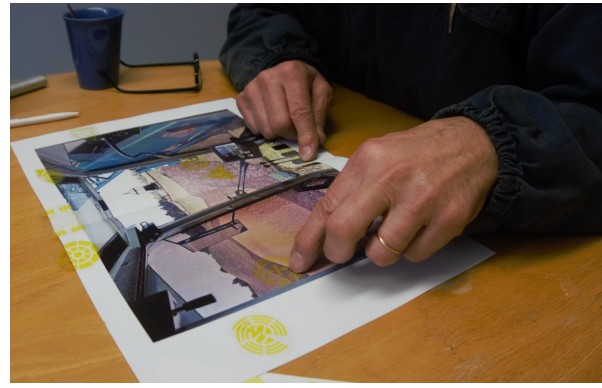

Figure 8: The operators were asked to place the proposed visualization concepts on the windshield according to their preferences. The image of the crane's cabin was downloaded from [16].

### 4.1 Proximity Warning

Both concepts for the proximity warning were made based on the top view of the mobile crane, with three levels of distance: near, medium, and far (see Figure 7a). In this study, we used humans as the form of obstacles for simplification purposes, and also because humans are moving objects. In practice, the obstacle can also be other things, such as buildings, trees, or overhead power lines. The visualization is always shown based on the direction where the cabin is facing.

In the first concept, there are two groups of segments and each group represents the presence of obstacle(s) on the left side or the right side of the cabin (see Figure 7b). The left segments will be turned on when there is an obstacle on the left side of the cabin,

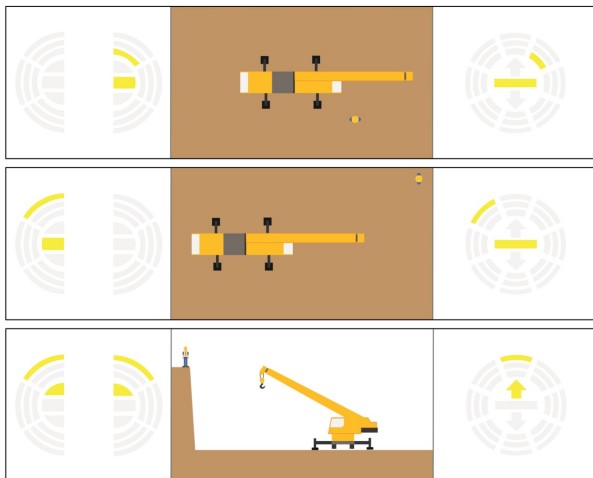

Figure 9: Some scenarios that illustrate how both concepts of proximity warning are used. The visualization is always shown based on the direction where the cabin is facing.

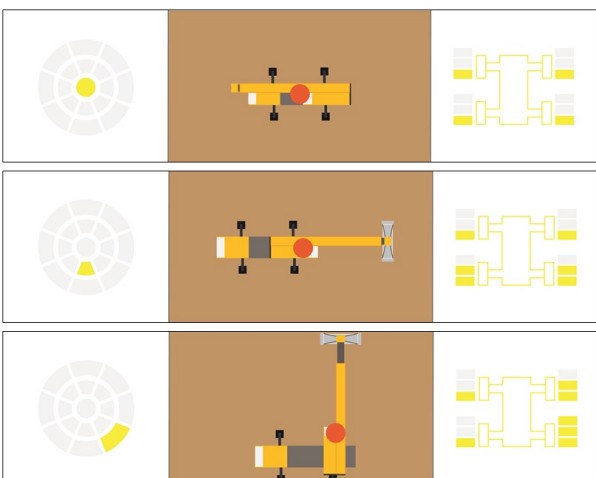

Figure 10: Some scenarios that depict the use of balance-related information based on the center of gravity (left) and loads on outriggers (right). The red circle represents where the center of gravity is, with respect to the machine. For both concepts, the visualizations are shown based on the direction of the front part of the machine, and not according to the cabin's direction.

and vice versa. As the visualization is split into two half circles, for obstacles that are exactly in the front or behind the machine, the same segment on both sides are turned on (see the bottom left image in Figure 9). The vertical segments show the position of the obstacle and its distance to the machine. The horizontal segments indicate three levels of altitude of the obstacle: lower, on the same level, or higher than the machine. The second concept is similar to the first concept, except that the visualization is in the form of a complete circle and the center parts indicate the altitude of the obstacle (see Figure 7c). See the images in Figure 9 for some examples on how the visualization will work in certain scenarios.

## 4.2 Balance-related Information

We have created two concepts which indicate the balance of the machine. The first concept is called 'center of gravity' and the second one is called 'loads on outriggers'. These names also suggest what kind of information being visualized.

The concept of center of gravity was also made based on the top view of the machine and it shows the current position of the center of gravity with respect to the center of the machine (see the left side images in Figure 10). When the center of gravity is near the center of the machine (the circle in the center), it shows that the machine is in a very stable position. To maintain the machine's balance, operators should ensure that the center of gravity does not go beyond the outermost segments, as the risk of tipping over is higher. Each segment in this concept indicates the position of the center of gravity.

The concept of loads on outriggers depicts the load that four outriggers have. Depending on the direction of the cabin and how far the boom is extended, each outrigger may have different loads. In this concept, there are three rectangles next to each outrigger. These rectangles are used to indicate three levels of load on each outrigger: low, medium, and high. The right side images in Figure 10 illustrate how this concept works in specific scenarios.

## 4.3 Wind Speed and Direction

In this concept, the arrows indicate the direction where the wind goes. In each direction, there are three arrows that indicate the force of the wind: low, medium, and strong. In the center, the segments indicate the estimated wind speed counted in kilometer per hour. See the images in Figure 11 for some scenarios that illustrate the use of this concept.

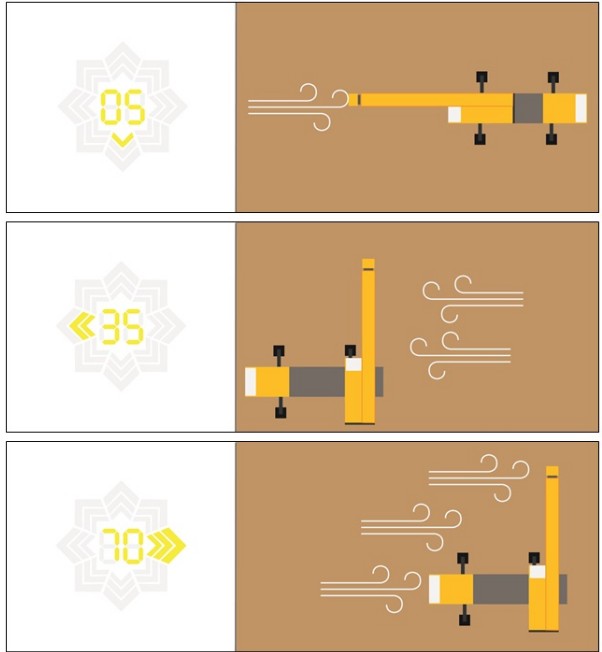

Figure 11: The arrows in the left side images and the wind icon in the right side images represent both wind direction and wind force. The wind direction is always shown depending on where the cabin is facing. The numbers in the center indicate the estimated wind speed.

## 4.4 Swinging of the Lifted Load

As the name implies, this concept indicates the swinging intensity of the lifted load. From the safety guidelines, we learned that the swinging could occur due to the wind, as well as the movement of the boom, and the swinging could affect the machine's balance. However, the visualization indicates the intensity of the swinging only, without telling the direction of the swinging. The reason behind this choice was due to the fact that the swinging could happen to any

direction, and thus could complicate the visualization. This concept shows something like a pendulum. The center segment is turned on when the lifted load is not swinging. The next two segments are turned on when the lifted load is swinging a bit, while the outermost segments are turned when the swinging is stronger. See the images in Figure 12 to see how this concept works.

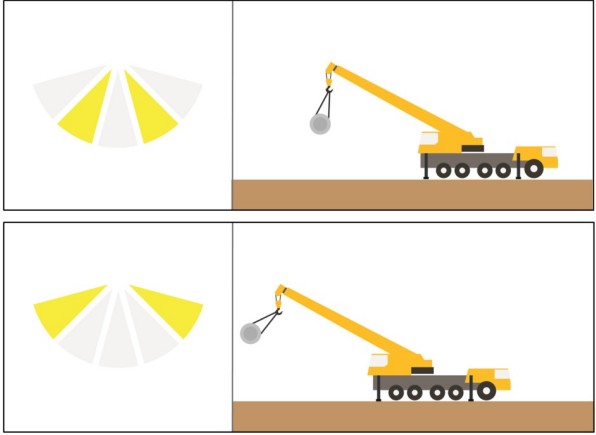

Figure 12: The images that illustrate how the concept works. If there is no swinging, the center segment will be turned on, while other segments will be turned off. Farther segments indicate stronger swinging.

## 4.5 Relative Load Capacity

Since the relative load capacity constantly changes depending on various factors [21], this concept shows four types of information: (1) angle of the boom, (2) height between the hook and the ground, (3) distance between the lifted load to the center of the machine, and (4) ten rectangles that each represents 10% relative load capacity (see Figure 13). The relative load capacity for each mobile crane, including the maximum limit for each influencing factor is usually documented and operators are advised to refer to that before performing lifting operations [15]. Exceeding the limit will cause the machine to tip over. In this case, the operators should prevent all rectangles from being turned on. See the images in Figure 14 for some examples that illustrate how the this concept works.

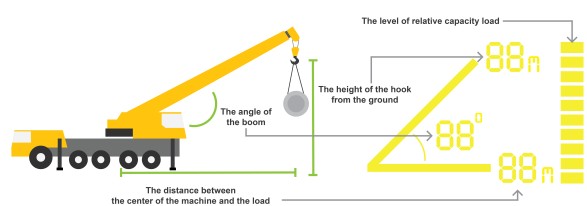

Figure 13: The meaning for each component in the concept for showing the relative load capacity.

## 4.6 Generic Warning Sign

The last concept was a generic warning sign that appears only when a collision or loss of balance is imminent to occur (see Figure 15). When this warning appears, operators should stop their current action.

## 5 RESULTS

This section presents the feedback on each visualization concept, as well as where the information should be placed on the windshield.

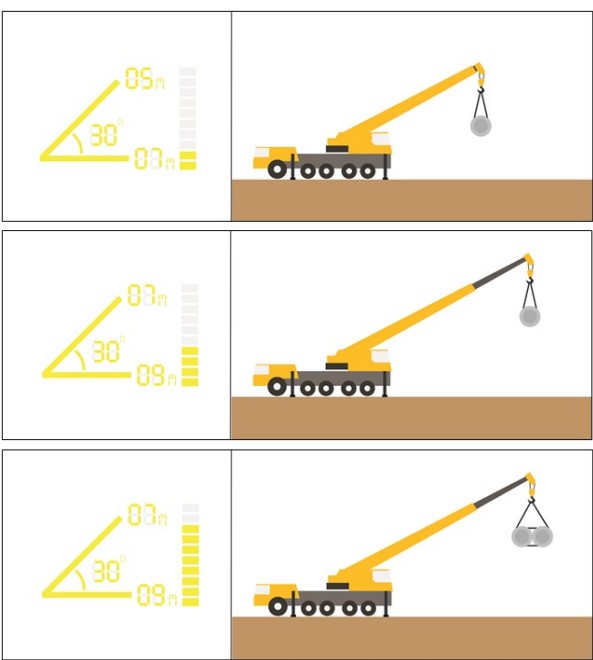

Figure 14: The images that illustrate how the concept works. Both top and center images illustrate that, even though the machine is lifting the same object, the relative load capacity varies depending on the height of between the hook and the ground, as well as the distance between the lifted load and the center of the machine. The bottom image depicts that the relative load capacity is of course increasing if the load is heavier. Note that the numbers in Figure 16 are used for simplification purposes only in order to demonstrate the concept.

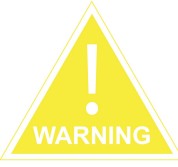

Figure 15: A generic warning that only appears when a hazardous situation is imminent to occur.

## 5.1 Feedback on Proximity Warning

When there was only one obstacle, it was quite easy for the operators to understand the meaning of both concepts and pinpoint the location of the obstacle. However, for the first concept, the idea of turning on the same segments on both sides, when something is exactly in front of or behind the cabin, was interpreted differently by the operators (see the images in Figure 16). The first concept was considered insufficient for all different scenarios, since if there are two different obstacles and have similar proximity on both sides, then the visualization will be the same as what is used for showing the obstacle that exists exactly in front of or behind the cabin. For the second concept, as the visualization is formed in a complete circle, it does not have the same drawback as the first concept (see Figure 16). The operators could easily pinpoint multiple obstacles using the second concept regardless of their proximity, and thus the operators preferred the second concept over the first one.

Furthermore, we also discovered that both concepts have another drawback for indicating multiple obstacles in different altitudes (see the images in Figure 17). In this case, it was not clear which obstacle is higher, on the same level, or lower than the machine.

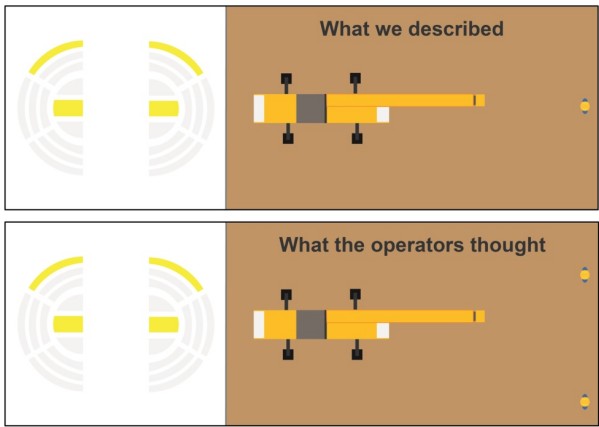

Figure 16: The first concept of proximity warning was understood differently by the operators when the obstacle is exactly in the front or behind the cabin. However, this way of thinking was not wrong either, since if there are two obstacles, where one is in the left side and the another one is on the right side of the cabin, the visualization will then look the same.

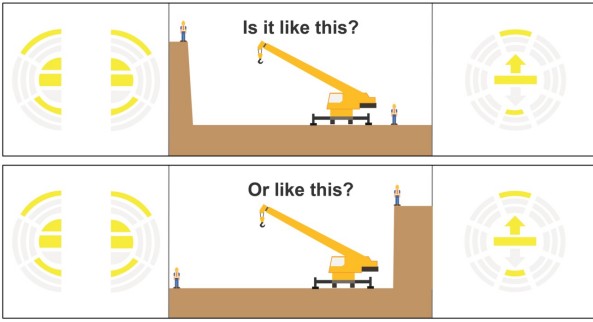

Figure 17: Both concepts are insufficient to visualize all different scenarios. In this case, although the operators were able to pinpoint both distance and position of the obstacles, the altitude of multiple obstacles could not be determined.

The operators also commented that it would be good if we can also show the indication of altitude directly in the segments that show the proximity of the obstacle. Despite this drawback, all the operators would like to have this kind of information on the windshield.

### 5.2 Feedback on Balance-related Information

Both concepts could be understood easily by the operators and there was no issue with the concepts. Only one operator preferred to have the concept of center of gravity, while five operators rated loads on outriggers as the better concept. The main reason was due to the fact that modern cranes already have similar visualization, thus they felt more familiar with it. However, only four out of six operators would like to have either concept presented in the cabin. The remaining two operators commented that this kind of information already exists on the head-down display, thus they felt that it is unnecessary to have it on the windshield as well.

### 5.3 Feedback on Wind Speed and Direction

The operators could easily comprehend the meaning of the concept, since modern mobile cranes already display something similar. Regarding the importance of having such information, the operators said that it highly depends on the weather. In a clear weather, this information is not needed, as the operators already know that the wind speed will be within acceptable limits. On the contrary, when operating the machine in other weather conditions, this information becomes critical for performing safe operations. Nonetheless, only three operators who would like to have this information all the time.

### 5.4 Feedback on Swinging of the Lifted Load

The meaning of this concept was obvious for the operators. However, only two operators who would like to have this information on the windshield. The remaining operators commented that this information is not needed, as they could see the swinging and estimate how the swinging will affect the machine's balance.

### 5.5 Feedback on Relative Load Capacity

This concept was also well understood by the operators, since modern mobile cranes are already equipped with LMI systems, which indicate similar information. According to all the operators, this is the most important information for performing safe operations and they would like to have it on the windshield as well. However, they commented that the information about the angle of the boom could be removed, as it is not important.

### 5.6 Feedback on Generic Warning Sign

Although the meaning of this concept was very obvious to the operators, only four out of six operators would like to have this warning shown on the windshield. The remaining two operators said that modern mobile cranes already have distracting auditory warning for imminent danger, thus the visual warning is no longer needed.

### 5.7 Information Placement

As mentioned in Subsection 3.3, we also asked the operators to place where the information should be visualized on the windshield. In this activity, they were also allowed to include or exclude some of the visualization concepts according to their preferences. Based on the placements that have been made by the operators, we can observe that there is a pattern on where the information should be presented (see the images in Figure 18). We can see that the operators would like the information to be visualized peripherally. They commented that the central area has to be clear from any obstruction, otherwise it is going to harm their operations. However, an exception was made by two operators who put the generic warning sign in the centre, since this position could attract their attention immediately. Regarding the placement of other visualization concepts, we unfortunately could not get a firm indication from this study, as the operators' preferences are quite diverse.

## 6 Discussion

This section describes the reflection on the evaluation method that we used in this study, as well as further suggestions that were given by the operators in the end of the interviews.

### 6.1 Reflection on the Evaluation Method Used in This Study

Since the visualisation concepts are proposals, and thus do not exist in their intended forms, we need to reason about their validity on the basis of the evaluation method that we used. Krippendorf [13] presents different levels of validity, in order of increasing strength, such as demonstrative validity, experimental validity, interpretative validity, methodological validity, and pragmatic validity. Due to the way this study was conducted, we are specifically discussing about demonstrative validity and methodological validity.

Regarding demonstrative validity, we were able to show the meaning of the proposed visualization concepts and how they could possibly work in different situations through the printed concepts on papers, along with the tools that the operators could interact with. This arrangement enabled the operators not only to understand the meaning of the proposed visualisation concepts more easily, but also having ideas on how the proposed concepts would work in various

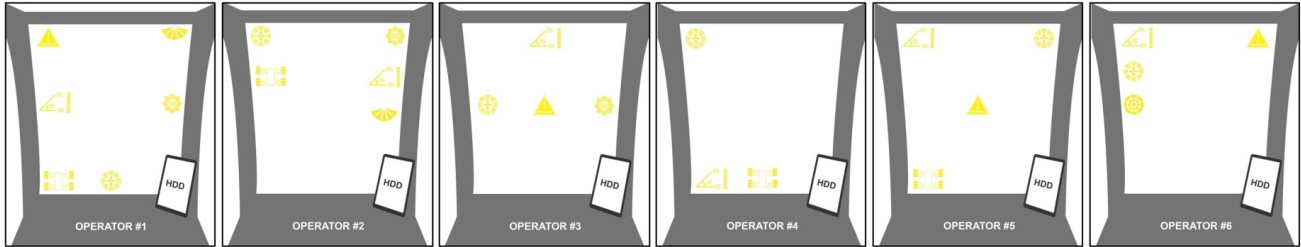

Figure 18: The images that illustrate which visualization concepts that the operators preferred to have and where the information should be placed on the windshield. Note that 'HDD' refers to the head-down display that already exists inside the cabin.

scenarios. In addition, we were also able to discover what would make sense or would not make sense according to the operators' way of thinking. For example, as what is presented in Subsection 5.1, we could discover that both concepts of proximity are inadequate for all different situations.

With respect to methodological validity, we decided to evaluate the proposed visualisation concepts in the paper form, since modifications could be incorporated easily in early stages. Despite using a low-fidelity prototype and some other tools, we were able to discover to what extent the proposed visualisation could suit the operators' needs and way of thinking in order to perform safe operations. Although the number of operators involved in this study is rather small, research on heavy machinery often involved small numbers of operators as the participants, either in observational studies [23] or experimental studies [24]. Our method was in contrast to what Kvalberg [14] has done, where the functional prototype has been developed, but there was no feedback from mobile crane operators. Needless to say, a prototype with higher fidelity that could be used in some scenarios, like what Chi et al. [7] and Fang et al. [11] have done, is required to determine to what extent the proposed visualization will benefit or hinder the operators.

## 6.2 Suggestions for the Proposed Visualisation Concepts

All the operators appreciated the effort of bringing the information closer to their line of sight. All of them agreed that this approach has the potential to improve safety in their operations, as they could acquire the information without diverting their attention from operational areas. Moreover, they also provided additional comments on how the transparent display could be made to better suit their needs.

Firstly, the operators raised concerns on how much the transparent display will obstruct their view in practice. As mobile cranes could be used in any time of the day, they concerned that the brightness of the transparent display may be too much for their eyes when the operation is done in dark environments. On the contrary, having a bright display will be good in bright environments, thus the information can still be visible even though there is a direct sunlight. Therefore, besides automatic adoption to the ambient light intensity, it is should be possible to manually adjust the transparent display's brightness.

Secondly, based on what is presented in Subsection 5.3, there were different opinions whether the information should always be presented on the windshield or not. Although the information is important, the information may not need to be visualised all the time. The operators also commented that it would be beneficial if they could choose what kind of information that will appear on the windshield, depending on their work environments. However, this kind of modification is not possible yet with the current transparent display, as the visualisation is fixed when the display is manufactured. However, if there are multiple transparent displays showing different kinds of information, it is should be possible to manually decide which transparent display that should be turned on or turned off.

## 7 Limitations and Future Work

The proposed visualization concepts presented in this paper were generated based on the findings from the safety guidelines review. According to the feedback in Subsections 5.2, 5.4, and 5.6, the operators could obtain similar information through looking directly at the environment or safety features that already exist on the head-down display, and thus having similar information presented on the windshield may not be so beneficial for them. Therefore, it is also important to take into account the availability of existing information inside mobile cranes and how the information is delivered in order to ensure that only essential information is presented on the windshield. However, we did not take this approach in this study, since different manufacturers may install diverse supportive information systems in their mobile cranes. As the result, one kind of information may be available in one mobile crane, but unavailable in another mobile crane.

In this study, we used a low-fidelity prototype, which was printed on papers, and some other tools to show the proposed visualisation concepts and to test the operators' understanding on the shown visualisation. With this arrangement, we were still able to convey the meaning of the proposed visualisation concepts to the operators, as well as to test their understanding with the help of the provided tools. Having said that, we are still limited in terms of the fidelity, and thus the results in this study are better to be considered as an indication.

In the future, we are planning to revise the proposed visualisation concepts according to the feedback from the operators that we have obtained. We are also planning to build a higher-fidelity prototype by building the actual transparent display that visualizes the proposed concepts. The prototype could then be used in future evaluations within controlled environments or real-world settings in order to investigate the impact of having such visualization on operators' performance in certain scenarios. For example, the number of things in the working environment that operators need to observe when operating the machine may influence the level of attention on the presented information. Furthermore, future evaluations could also be carried out to discover which placement of information that provides the optimum result for the operators.

## 8 Conclusion

In this paper, we have proposed and evaluated the visualization concepts using transparent displays that could be used to assist mobile crane operators to perform safe operations. We started the design process by gathering information from few safety guidelines, generating ideas by conducting a design workshop, and then obtaining feedback from the operators through interviews. The operators that we have interviewed appreciated this approach and the results from this study indicate what kind of information that operators need in order to perform safe operations, how we should visualize the information, and where to place the information on the windshield.

Nonetheless, more studies, such as evaluations with some scenarios using high-fidelity prototypes, will need to be conducted to further determine both applicability and usefulness of this approach.

## ACKNOWLEDGMENTS

This research has received funding from the European Union's Horizon 2020 research and innovation programme under the Marie Skłodowska-Curie grant agreement number 764951.

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
