# OpenReview forum: "Presenting Information Closer to Mobile Crane Operators’ Line of Sight: Designing and Evaluating Visualization Concepts Based on Transparent Displays"
_graphicsinterface.org/Graphics_Interface/2020/Conference — GI 2020_

### Official Review · AnonReviewer2 · 2020-04-20
**This paper presents the design and evaluation of visualization concepts using transparent displays to assist mobile crane operators to perform safe operations. The research questions are well motivated and answered.**

**Rating:** 7
**Confidence:** 4

**Review:**

This paper presents the design and evaluation of visualization concepts using transparent displays to assist mobile crane operators to perform safe operations. The contribution of the work is the visualization concepts and the feedback of the visualization concepts from mobile crane operators.

The three research questions are well motivated and logically connected. The method used is appropriate to generate low-fidelity paper prototypes of visualization concepts and to understand how the concepts were understood and could be improved with target users---mobile crane operators. The details of the method are well presented and the illustrations are helpful for readers to comprehend the method. The paper is well written and easy to follow.

That said, the paper does have limitations. The visualization concepts were only tested with a low-fidelity prototype and the focus was more on the understandability of the concepts. However, there remain questions to explore. For example, the contexts of operation tasks are not considered in all tasks. Operating a mobile crane on a typical construction site is different from operating it on a city road. The types and intensity of information that mobile crane workers need to attend to and take care of are different. Such contextual challenges are hard to simulate and construct with low-fidelity paper prototypes, which are known to be limited in their interactivity.  It is necessary to acknowledge this limitation and other similar context-induced challenges that the current method is unable to handle.

In terms of visualization concepts, the visualizations of swinging of the lifted load (figure 14) have only considered the swinging degree but not the direction of swinging. Swinging back and forth can cause a different effect on a mobile crane than swinging left and right. Was the swinging direction intentionally left out in the visualizations? If yes, what was the rationale?

It is interesting that mobile crane workers commented that some visualized information was not necessary simply because they could directly observe such information (e.g., swinging of the lifted load) or such information has already been shown (e.g., on head-down display). This makes me wonder there are other types of visual information (e.g., directly observable information and information on the head-down display) that must be taken into consideration when designing the visualization concepts. This point should be discussed in the paper.

In sum, the submission makes a valid contribution. With the comments incorporated in the revision, I would feel confident to support the acceptance of the submission.

---

### Official Review · AnonReviewer3 · 2020-04-21
**Not strong contribution**

**Rating:** 4
**Confidence:** 3

**Review:**

The paper is well-motivated, but I do not think this paper presents sufficient contributions for acceptance.

First of all, the contributions are weak.
The formative study and the low-fi design prototype may not be strong enough for this conference.
I understand the actual deployment is challenging, as there are many safety concerns, but still, the authors could develop and evaluate the proposed design through VR head-set with 360 photos of construction site + visualization, or something similar.
It is also uncertain how this visualization can be integrated with the other sensor values, such as wind, proximity sensing, etc. I know the authors argue this is beyond the scope of the paper, but in that case, the contributions seem a bit weak.

Second, the authors did not sufficiently review the literature.
There are tons of work of see-through display or projection mapping for information visualization or instructions. For example, from the classic one like Feiner's "Knowledge-based Augmented Reality" which has over 1300 citations, to relatively recent ones like Willet's "Embedded Data Representations".
The authors should review these prior works that explore visualizations to increase awareness with augmented reality.

There are also many missing works in the immersive environment for safety purposes. Even limited in this specific application of crane operations, I was able to quickly find:
- Development of user interface for teleoperated cranes
- Attention-based user interface design for a teleoperated crane
- Using affective human–machine interface to increase the operation performance in virtual construction crane training system: A novel approach
- SimCrane 3D +: A crane simulator with kinesthetic and stereoscopic vision
- Multiuser virtual safety training system for tower crane dismantlement

Although I am not an expert in this domain (AR/VR for safety applications), it sounds like the literature review of this paper may miss some important prior works.

There are also some minor points, such as the presentation of the work can be improved (e.g., Figure 4 is almost unreadable) or concerns about how the findings of the paper can be generalizable beyond this specific application (given the author's claim about the design guidelines is the main contribution), but my major concern is the first point (i.e., the significance of the contribution).
Therefore, I would not recommend for acceptance.

---

### Official Review · AnonReviewer1 · 2020-04-21
**Interesting first step towards addressing information dispersion in mobile cranes**

**Rating:** 6
**Confidence:** 4

**Review:**

The authors have prototyped potential transparent display based solution to bringing critical information closer to the person operating a mobile crane. The authors identified a problem of information disconnect wherein the operators are looking out through the glass to move the crane and all the supplementary information is presented on a screen in the bottom corner of their field of view. The proposed solution identifies critical information and surfaces it in the field of view by leveraging transparent displays, so as to minimize any obstructions to the view. Three HCI researchers designed various visualizations to present the critical information, the authors created low-fidelity prototypes to conduct initial testing of the usefulness of the visualization, and recruited six crane operators to undergo an interactive session where they evaluated the different visualizations and its utility in their own workflow. The paper presents an interesting new domain to the HCI community and provides an initial study toward what can be a longer term research project aimed at building out such critical, real-time systems.

Although the paper was written well-enough to comprehend, I found the sectioning of the content to be non-traditional. Few examples:
- Research questions which would ideally be mentioned near the end of introduction section were first introduced in the methods section.
- Parts of method and design of the study was mentioned in the results section. I would ideally prefer to see the system design and rationale before diving into the results.
I would point the authors to Professor Wobbrock's guide on writing HCI research papers for structural guidelines which I am talking about[1].

While I appreciated reading about the study protocol which was something different and new for me, there were some limitations which could have been addressed to further improve the paper.
- Figure 7. I understand developing working prototype was difficult. However, it would have been easy to print a post with the view that is being shown to the operator to get a 1:1 ratio of the view and the displays. In the current set-up the operator has to do additional mental calculation of scaling the displays for potential use in their workflow.
- Some of the findings and discussion around preferences or what information operators would like to see and where, could have been gathered earlier through some formative interviews or surveys.

Some additional comments to further improve the paper:
- Figure 4 needs to be redone. Increase the contrast even if it reduces the quality of the image, as currently, the image is quite illegible. You may even consider shortlisting 3-4 of these images and enlarging them for added clarity.
- Section 3.1 In coming up with the list of information which is critical, was any expert consulted? Is any member of the research team and expert in the area? If so, highlighting that may alleviate any concerns regarding the validity of the choice of parameters.
- Section 3.2 Are the three researchers mentioned here also the authors of this paper? If so, call it out for added clarity.
- Section 3.2 Given the heavy restrictions on transparent displays (only available in two colours, can only present static visualizations) why did the authors not consider other alternatives such as AR glasses (e.g., Google Glass, Microsoft Hololens). A rationale of this would be helpful to have in the paper.
- Table 1, unless information of age and experience is highly critical, I would recommend presenting them as ranges so as to further obfuscate the participants and avoid any unnecessary identification. (e.g., age between 30-40, experience between 10-15 years)
- Section 3.3 It was not clear what authors meant by "forms" on page 4. (There were ten forms for each concept ...)

Overall, even though I point of several ways in which the current submissions could have been stronger, the paper does have merits. It presents a novel problem area and potential ways of addressing them. As such, I do not see a strong reason to reject the paper in its current form. There are several small changes I suggested above which the authors can make in the camera ready submission to improve it.

Reference:
1- Catchy Titles Are Good: But Avoid Being Cute. Jacob O. Wobbrock

---

### Meta-Review · Area_Chair1 · 2020-04-21

**Recommendation:** Accept
**Confidence:** 4

**Metareview:**

R1 and R2 agree that the paper has merits and can be improved to be well-above the acceptance threshold with small modifications. R3 brings up several good critiques for the work. I would recommend reading their feedback and incorporating the changes in the related work section.

R1 and R3 both bring up the relevant area of AR and how that can not just enhance but bypass certain restrictions of transparent displays. Maybe adding a discussion around that point would be fruitful to have.

R2 brings up some good questions for the visualization design rationales. Consider addressing them.

---

### Decision · Program_Chairs · 2020-04-25

Accept